# An Economic Alternative from the Base: The Basic Ecclesial Communities (BECs) in the Diocese of Boac, Philippines

**Willard Enrique R. Macaraan** 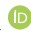

Department of Theology and Religious Education, De La Salle University, Manila 1004, Philippines; willard.macaraan@dlsu.edu.ph

**Abstract:** Roughly dialogical in approach, this paper attempts to converse the ideation of Jesus' narrative of compassion as embodied in biblical literature with the praxis of compassion as exemplified in the socio-economic dynamics of selected Basic Ecclesial Communities (BECs) in the Diocese of Boac, Marinduque, Philippines, namely the communities of Panag Kalangkang, Bahi, and Dolores. Its purpose is to propose a compassion-based economic alternative that incorporates the salient features of Jesus' compassion as its thematic anima (soul) and the praxeological attributes of BECs, particularly their integration of community stores, as its ideal corpus (body). This paper argues that as an ecclesial unit, often and traditionally constrained to liturgical and spiritual performances of faith, the BEC can be a suitable base to host and embody an economic dimension that features ethical and moral dynamics responsive to the call of Pope Francis for a new economic model.

**Keywords:** Albert Nolan; economic model; compassion human economy; Marinduque; cultural revolution

## 1. Introduction

Situated within the theme of critical analysis of mainstream (neoclassical/neoliberal/laissez-faire) economics, this paper attempts to contribute to the project of "compassion-based economics". In his apostolic exhortation *Evangelii Gaudium* (Francis 2013) and encyclical *Laudato Si* (Francis 2015), Pope Francis has laid out his strong attack on the current state of the global market that endangers human persons to commodification and results in what he refers to as a "throwaway" culture (*EG* 53). He adds that, " . . . we end up being incapable of feeling compassion at the outcry of the poor . . . the culture of prosperity deadens us" (*EG* 54). It is then urgent for a committed response that involves a "bold cultural revolution . . . to recover the values and the great goals swept by our unrestrained delusions of grandeur" (*LS* 114) and "to the return of economics and finance to an ethical approach which favours [sic] human beings" (*EG* 58).

Within this call for a 'cultural revolution' in the market system is this paper's attempt to reinforce the value of looking into an economic alternative that orients itself to interpersonal relations rather than to profit gains. From economics that is based on interpreting data and valuations based on charts, graphs, and numbers, the paper explores economics that zeroes in on the vision of flourishing and intersubjectivity of the people within market dynamics. On this note, the paper attempts to nuance "compassion" not just as a secular value but as a praxis that is demonstrated within Jesus' Kingdom proclamation. The paper, therefore, inquires into the meaning and dynamics of economics that are founded on the ideological nuances of Jesus' praxis of compassion. As suggested by Boff (1987) in his "dialectic hermeneutic" that any "reference to scripture should be by way of *creative* memory, and the readings of scripture should be a *productive* reading," this paper attempts to draw salient features of the praxis of compassion based on an exegetical survey of the Gospel text. In response to the call of Pope Francis (2019) "to set in place a new economic model", this paper hopes that a compassion-based economics may be a feasible alternative to mainstream neoliberal economics.

Compassion-based economics is herein understood as a non-standard economic alternative, which does not only intend to serve as a critical challenge to the rationalized *ethos* of the dominant system but also as a genuine model of people-centered economics. In the same mold as that of solidarity economies (e.g., communal bank in Venezuela), gift economies (e.g., "dama" of Mali, West Africa), and sharing economies (e.g., economy of communion by Focolare), compassion-based economics rests on the theorization that markets are believed to be social creations in which cultural elements influence economic processes and as a cultural construction, the market reinforces the power of the social agents in the culturalization of economics (or "economization" of culture). Veering away from highly-mathematized-rationalized-calculations of modern economics and system theories, (neo)substantivist theorization on economics has provided strong theoretical support for the feasibility of a person-oriented economic behavior/*ethos*. For Polanyi ([1944] 2001), "a substantive meaning of economics is simply the way the society needs, understood in the broader sense of household 'provisioning' and not much on the maximization of utility". It is a great contrast to the formalist understanding of economics which is determined by the logic of rationalized decisions amidst scarcity of resources. It is argued that non-economic values, motivations, and social relations are largely constitutive of the entire market system. Aside from the more obvious neoliberal objectives of profit, utility, and material gains, the marketplace is also a space where kinship, friendship, and other intangibles are circumscribed within a (rational) decision in a marketplace.

In the project of compassion-based economics, the author proposes to integrate the socio-cultural components (*corpus*) embodied in BECs and the ethical-moral valuations (*anima*) contained in Jesus' praxis of compassion. Metaphorically, the *corpus* provides the structure while the *anima* is the spirit that permeates compassion-based economics. A BEC is a localized entity and is essentially small in size and scale. It is decentralized; formed from the ground; and is oriented towards it. Its embedded culture is based on the real encounter of persons who know one another. Since there is strong interpersonal interaction among members of BECs, an integration of socio-economic artifacts or activity is generally less concerned with profit than the well-being of each member of the neighborhood community. To explore more nuances of the structural components of BECs, the author performs fieldwork on three (3) BECs under the Diocese of Boac, located in the province of Marinduque, Philippines. As for the articulation of salient features that would provide for the *anima* of the economic alternative, the author intends to look for "orientations, models, inspiration, and principles" (Boff 1987) from Jesus' narrative of compassion through an exegetical survey provided primarily by Albert Nolan (2008) whose book, *Jesus Before Christianity,* provides a historical-critical explication of the praxis of Jesus' compassion as viewed from the Gospel text.

## 2. Jesus' Narrative of Compassion

Using primarily Nolan's historical-critical exegesis, this section shall briefly and generally discuss Jesus' praxis of compassion as found in the Gospels and attempt to highlight three (3) salient features for the *anima* of a compassion-based economic alternative. The compassion of Jesus is his response to suffering; a commitment "to liberate people from every form of suffering and anguish—present and future" (Nolan 2008, p. 36). The degree and intensity of the undeserved suffering of the poor in the time of Jesus go beyond the physical and physiological pain of hunger and disease as it is the emotional suffering that they have to endure in their daily struggle to merely survive. It is their resignation to hopelessness and absence of a way out of such an ordeal that everyday life has become burdensome and meaningless. It is, in a sense, a kind of existence where they survive not 'to live' but merely not 'to die'.

Jesus' response in the midst of this is compassion which is not a mere feeling of empathy or pity but is strongly characterized by willful determination to seek the victim's liberation. As most of his stories of healing would end with a message of peace, "Go in peace", his narratives of forgiving sinners that are usually situated within a table fellowship

ended with the same call to live in peace as well and a commitment from both the healed and forgiven to convert and change their ways. For them, it was a moment of total liberation from the past—gratuitously and unconditionally (Nolan 2008, p. 50).

Jesus' praxis of compassion was a confluence of empowering the individual's lost sense of self-worth and at the same time a critical stance against oppressive political structures and religious traditions. Jesus' stance and position were generally marked with open defiance against the structural systems, both political and religious, where the lives of the many were defined and determined by the whims and caprices of the powerful few. Jesus' response of immediate 'relief' to the poor does not fully define his project of total liberation. Authentic liberation urged not only a mere conversion of the heart of individuals (*metanoia*) but also an audacious moral-cultural reconstruction of political and religious (super)structures.

The character of Jesus' compassion to "be willing to suffer until the point of death" is considered by Nolan to be a paradox where "the person who fears death is already dead, whereas the person who has ceased to fear death has at that moment begun to live. A life that is genuine and worthwhile is only possible once one is willing to die" (Nolan 2008, p. 139). In the garden of Gethsemane, Jesus had shown how he struggled with the thought of his impending death. It was an unimaginable agony, to say the least. There was that moment he could not discern the will of His Father but submitted to His will anyway. "My Father, if this cannot pass unless I drink it, your will be done" (Mt. 26:42).

Authentic compassion extends to losing one's whole self (prestige, possessions, family, power, and even life), all for the sake of others. Compassion compels people to do everything for others. It is therefore absurd to claim a life lived *for* others if one is not willing to suffer and die *for* them. "... [H]e humbled himself, and became obedient unto death, even death of the cross" (Phil. 2:8).

The basic attitude of Jesus' compassion was solidarity with humankind that took precedence over every other kind of love and solidarity, "If anyone comes to me without hating his father and mother, wife and children, brothers and sisters, and even his own life, he cannot be my disciple" (Lk. 14:26). What Jesus was merely asking for was "detachment"— of not giving preference to one's family and relatives. It was a total sacrificial stance to disregard the kinship identification and treat friends and members of one's family with the same categorical identification with the rest of the victims—as human persons.

### 2.1. Nuancing the Linguistic Features of Compassion in Scriptures

The original Greek word used in the Bible, σπλαγχνιζομαι [*splanchnizomai*] which is roughly translated in English as "compassion" refers to the inward parts, bowels, entrails, intestines, the viscera, the guts, more particularly the *viscera thoracic* (i.e., heart, lungs, liver, kidneys). Philologically, therefore, the concept of biblical compassion is rooted in literal body parts, located inclusively in the abdominal and thoracic areas (heart, lungs, liver, intestines, bladder, kidneys, womb, etc.).

In a lexicon edited by Liddell et al. (1940), conventionally referred to as *LSJ*, the anatomical reference associated by ancient Greek poets and writers such as Aeschylus, Pindar, and Sophocles with σπλαγχνον [*splanchnon*] was a locus of more violent passions and emotions, such as anger and love. In contrast, the ancient Hebrews referred to the same word as the viscera were the seat of the more tender affections, kindness, benevolence, and compassion (Thayer [1841] 1889). Eventually, the body parts became equivalent to the feelings themselves. Wigram mentions four instances where the idea of compassion has been used in the New Testament. The first was εςλεεω/εςλεαω [*eleeo/eleao*], a verb found in the Synoptics, in the letters of Paul [Rom. 9:15, 18; 11:30ff; 12:8; 1 Cor. 7:25; 2 Cor. 4:1; Phil. 2:27; 1 Tim. 1:13–16] and in the Catholic epistles [1 Pet. 2:10; Jude 1:22] (Wigram [1839] 1970). Most modern versions refer to it as "mercy" or "pity" and this is what the people seek and beg from Jesus during narratives of healing and exorcism [Mk. 10:47ff; Mt. 9:27, 20; Lk. 18:38; Mt. 15:22; Lk. 17:13]. The second was the verb συμπαθεω [*sympatheo*] and adjective συμπαθηνς, [*sympathes*], which refer to sharing "the same feeling

of suffering" (Heb. 4:15; 10:34 and 1 Pet. 3:8). The third was the verb οικτρω/οικτειρω [*oiktiro/oikteiro*] which conveys a sense of "pity, a feeling of distress through the ills of others" [Rom. 9:15; 12:1–2; 2 Cor. 1:3; Phil 2:1; Col 3:12; Heb. 10:28]. The fourth was the most recurrent word in the entire New Testament. The Greek verb σπλαγχνιζομαι [*splanchnizomai*] is found 12 times in the Synoptics. Its root word σπλαγχνον [*splanchnon*] is found 11 times in the New Testament—twice in the Lukan corpus, once in the Johannine literature, and the rest are in the letters of Paul. The subject of this paper is this fourth instance of biblical compassion.

*2.2. Salient Features*

To attempt historical objectivity on Jesus' compassion praxis is to a certain degree always imperfect. Every historical presentation is always written and told from a certain context, perspective, and worldview. This paper draws its exegetical survey from Nolan's (2008) "*Jesus Before Christianity*" due to how he stresses the "context" of Jesus' socio-politico-religious world which is crucial in any attempt to interpret the text of the Gospels especially if there is an intention to relate that world of Jesus to today's world where people somehow suffer the same problems although in a different degree. For Fox (2012), Nolan's desire to situate the gospel in that cultural setting provides a framework to look for connections, clues, and contradictions that may shed light on Jesus in his human skin (Fox 2012). Among all of the many ideations that Nolan's study has revealed from a historical-critical lens, it is his compassion for the suffering, the poor, and all those rendered powerless by their socio-economic status that is a defining characteristic of his mission (Fox 2012). To achieve what Boff refers to as hermeneutic competency, the exegetical survey would look for "orientation, models, types, directives, principles, inspirations—elements permitting us to acquire . . . the capacity to judge... the new, unpredictable situations with which we are continually confronted" (Boff 1987, p. 149). On this note, the paper lists three salient features of Jesus' compassion based on Nolan's exegetical description and will be discussed in detail from hereon.

2.2.1. Jesus Identifies Himself with the Poor and Outcast in Friendship and Solidarity (Option for the Poor)

A victim's experience of Jesus' compassion was an enigma of the sort that engaged the victim to forever tie one's life to Jesus, just as a disciple to his master or a lost sheep to his shepherd. Conversely, it was Jesus' compassion that moved him to build a friendship with the victims just as it was the same experience of Jesus' compassion that makes the victims respond to such an invitation of solidarity and communion. This mutual direction allowed both Jesus and the victims to forge ties and friendships that may perhaps be mistaken as another form of group solidarity, which the Jews at the time of Jesus' society were so fascinated with forming. Derrett notes that after prestige and money, the fundamental concern of the society in which Jesus lived was group solidarity (Derrett 1974).

Compassion is Jesus' response to the suffering of people. It leads him to a commitment to action "to liberate people from every form of suffering and anguish—present and future" (Nolan 2008, p. 36). The cure of the leper (Mk. 1:41), of the two blind men (Mt. 20:34), and the rest of the healing narratives of Jesus ended with a message of peace—contentment, happiness, an authentic sense of liberation from the chains that shackled the victims to live at the margins. A similar experience of relief and gratitude was felt by the sinners where after dining with Jesus in a table fellowship, it ended with the same call to live in peace (Lk. 7:48, 50).

As Jesus was moved with compassion by the state of dehumanizing conditions of the victims, he hoped for nothing else but their liberation and redemption from such painful strife. It was essentially a 'good' and 'true' conviction that something can and will happen since it was good and since it was true that goodness can and always triumph over evil. Jesus' compassion is his response to their suffering and his conviction of goodness, of their liberation from pain and distress, resulting in nothing else but salvation, only since it is

a good conviction. "The power of faith is the power of goodness and truth, which is the power of God".

It is clear therefore that for Jesus, his praxis of compassion awakened and strengthened the faith of the victims. More than that, his compassionate behavior resulted in miracles of healing, forgiveness, and solidarity since essentially part of it was the strong conviction toward good demonstrated by both Jesus and the victims. The intentional nature of Jesus' compassion to bring about the good as conditioned by a strong faith in God and oneself was the platform by which these huge stories and narratives of miracles took place. One can therefore assume that from the foregoing, the praxis of Jesus' compassion necessitates strong fiducial quality to actualize the 'sentiment' of 'feeling-with-the-other-in-pain'.

### 2.2.2. Jesus' Compassion Is Inclusive Inviting All to Friendship and Community Based on Humanity and Personhood (Personalist/Intersubjective/Communion Orientation)

One of the powerful images of Jesus' passion for inclusivity was his table fellowship. Jean Anthelme Brillat-Savarin, a famous French gastronome known for his adage, "Tell me what you eat and I will tell you who you are", implies the close relationship between the person and the food he consumes. But for Jesus, it is more on who he eats the food with that defines who he is. In the time of Jesus, meal gathering was one of those social events where ranks and status were established. The social ordering started with who were invited and where they were arranged in and around the table. For Smith, "Such distinctions and honors were considered essential to the makeup of cultured society, and the banquet normally functioned within society to buttress its view of status" (Smith 2003).

A problem may emerge from an unwitting eye though since "... the Pharisees and the scribes began to complain, saying, 'This man welcomes sinners and eats with them'" (Lk. 15:2). The scandal of Jesus' inclusive banquet is epitomized by his table fellowship with the outcasts of society and his eating with them as a friend (McFague n.d.). While it was true that Jesus customarily dines with tax collectors, sinners, and even the beggars and tramps, he must also have invited the Pharisees and the well-reputed ones. The intention of an open meal invitation was there to start with. His inclusive banquet does not exclude the rich and those who enjoy social prestige and privilege since it is a universal invitation (Nolan 2008). The problem was the way these people responded. The Pharisees and the 'respectable' ones may have second thoughts about participating in Jesus' meal where they can be seen sharing the same table and food with the social outcasts. It was more of the fear to lose status by accepting Jesus' invitation. No wonder, the parable of the Great Feast in Lk. 14:15–24 was perhaps a reflection of the experience of Jesus when his invitation was turned down by a lot of them.

Jesus' willingness to eat with sinners (Mk. 2:15–17; Mt. 11:19; Lk. 7:34,39; 15:1–2; 19:7) finds semantic parallelism with Paul's confrontation with Peter over the latter's 'separation' from table fellowship with 'Gentile sinners' (Gal. 2:12, 14–15). For Dunn, it may well be deliberate for Paul to allude to Jesus' tradition of table fellowship with sinners for Peter to recognize the allusion and that Peter would be shamed accordingly (Dunn 1998, p. 192).

Jesus' fundamental regard for universal solidarity, his negation of group formations at the expense of marginalizing others, rests on the core idea of treating the person as a person. The perspective of humanness as the criterion for treating others, regardless of capability and have-ability, paved the way for a creative formation of communion among people across various social differences. From a compassionate stance of aiding the suffering other to Jesus' blanket invitation for communal solidarity as human persons, one cannot deny how Jesus' anthropology highlights that the fullness of the human person as an individual is total and accomplished in its integration to communion with the rest of humankind. The sense to belong oneself to a solidarity-based communion is an aspect of Jesus' praxis.

### 2.2.3. Jesus' Compassion Empowers the People towards Total Human Liberation (Liberationist-Enabling Direction)

Political liberation from the hands of Roman imperialism was a recurring theme in Jesus' Palestine. The promise of liberation was dominant in the text of the prophet

Isaiah (29:18–19; 35:5–6; 61:1–2) and its reiteration in Lukan text (4:16–21) was seen by experts as Luke's way of pushing the thesis of Jesus' messianic function in the context of 'expecting' Israelites. What was rather bold in Jesus' praxis was his open defiance against the structural systems, both political and religious, where the lives of the many were defined and determined by the whims and caprices of the powerful few. Compassion's bold trait of unrestrained resolve to effect liberation and authenticity is warranted by the genuine misery of the afflicted and cursed. In short, what can be guaranteed by the spirit of compassion is its unwavering decision to act in favor of the oppressed despite unforgiving and unsupportive structural conditions.

Jesus' intention is clear: it is a call towards conversion first of individual character, attitude, ethos, and inner soul. He does not resort to battling the Romans in war as it is not only immoral but also impractical. Riddled with internal division, conflict, and social fragmentation, Israel does not stand a chance against the military might of the Romans. "Jesus prophesies an unparalleled military defeat for Israel" (Nolan 2008, p. 22). Jesus' proposed way was neither a matter of blind and passive submission to Roman oppression nor a violent uprising against it. The case was to reach down to the root cause of all oppression and domination: the lack and absence of compassion in humankind. Nolan argues that Jesus' bottom-up approach can eventually affect social unity and cohesion which in the eyes of the Romans may be enough reason for them to withdraw their hold of Israel and eventually release them to freedom and independence (Nolan 2008). It must start by recognizing the small discourses and narratives of those at the margins and reintegrating them into society with privileges, rights, and duties accorded to any citizen of society.

A summary of these three salient features is herein summarized in Table 1.

**Table 1.** Jesus' praxis of compassion.

| Salient Features | Description | Elements of Action |
|---|---|---|
| Option for the poor | Jesus identifies himself with the poor and outcast in friendship and solidarity. | Identification with the margins, identity evaluation, empathy and sympathy to suffering ones |
| Personalist/intersubjective/ communal orientation | While there is a preference for the plight of the poor, Jesus invites all to friendship and community on the basis not of prestige, wealth, and status, but humanity and personhood. | Friendship, community-building, social equality, person-orientedness, humane conditions |
| Liberationist-enabling direction | Jesus empowers the people of Israel to a change of heart or *metanoia* as an initial step towards political liberation. | Empowerment, enabling structures and systems, capacity building |

## 3. The Praxis of BECs: Integrating a Community Store

The fieldwork is located in the Diocese of Boac, in the island province of Marinduque. Surrounded by water, the main access of entry is through the seaport and is about 210 km. off Manila, the capital of the Philippines. It has a population of around 234,521 as of the 2015 demographic census with about 88% of them professing the Catholic faith (Philippine Statistics Authority 2018). Their main source of livelihood is agriculture and fishery. According to the Catholic Bishops Conference Episcopal Committee on Basic Ecclesial Communities (CBCP-BEC 2013), the Diocese of Boac has 14 parishes and since 1982, its central pastoral thrust has been organizing Basic Christian Communities or locally termed as *Batayang Pamayanang Kristiyano* (BPK).

In recent years, Boac's BPK program has been considered an outstanding BEC model at the national level mainly due to its holistic orientation with strong integration of a

social-action component aside from liturgical-evangelical components. One of their notable projects is the Marinduque Social Action Multi-Purpose Cooperative (MASAMCO) community store. which is one of the flagship projects under the Diocesan Social Action Center (SAC). While most of the BPKs in Boac has integrated socio-economic components like candle-making, worm composting, and crispy *malunggay* (horseradish) chips among others, it was the integration of a community store in their BPKs that is selected as the subject of field research since most of the BPKs have it due to fairly easier technical skills requirement and lower maintenance cost. In choosing which BPK community store to concentrate on, the SAC recommends three BPKs: *Panag Kalangkang*, *Dolores*, and *Bahi*. While all of these community stores are generally guided, oriented, and regulated by SAC, it is interesting to note that their geographical, demographical, and social dynamics and context show variations and nuances that eventually impact the success or failure of the community store in their respective BPKs.

In its fieldwork, the author had visited the diocese on two separate occasions with personal interviews with the bishop and some of the priests of the diocese. The author also had initial meetings and orientations with the officers and stakeholders of the SAC before personally meeting the community members of different BPKs. The conversation with the members of the BPKs was informal and specific to how the community store is maintained and its impact on their communal dynamics. For both *Bahi* and *Dolores*, the small group gatherings were each held inside its own BPK chapel while for *Panag Kalangkang*, it was held in an open space in front of the community store since the community did not have a chapel.

### 3.1. The Three (3) BPKs

*Panag Kalangkang* is a small fishing village situated in an islet off the coast of Sta. Cruz, Marinduque, about thirty kilometers away from the capital city of Boac. It has for many years become a self-reliant BPK due to its self-sustaining status and successful profit-sharing mechanism where additional income significantly augments peoples' daily needs (Macaraan 2016, p. 88). As a fishing community, the sea is its main source of livelihood. Sustenance is affected when the weather is bad and fishing is not an option. But with the community store, the members found a way to get by as it offers an alternative source of income and immediate resources, especially in times of weather disturbances.

This store has expanded its commercial value since it has also become a physical landmark for a social gathering where community members share stories, discuss plans, and resolve conflicts. It has become a stomping ground where they hang out. It has become a communal artifact and a central piece in their life as a community. When the store caught fire a few years ago, they cooperatively helped put out the fire and together rebuilt it. For them, it was not just a store they were saving; it was their life, their community (Macaraan 2016, p. 89).

*Dolores* is a rural village located within the town of Sta. Cruz, Marinduque. It consists of about 21 families whose members are either coconut farmers or professionals employed in various institutions and fields. Their community store was organized years after *Panag Kalangkang's* and what the members wanted then was to somehow replicate the success of *Panag Kalangkang*. However, it has failed to achieve the same success. Instead of unifying the members together and providing a source of a harmonious relationship among them, the store has been viewed at times as a source of social conflict and personal/familial burden. One of the main objectives of a MASAMCO community store is to instill a sense of communal responsibility in each of its members. Instead of hiring a paid individual to look after the daily operation of the store, each member or at least a member from each family is assigned to handle the daily operation of the store. There are times when some members fail to religiously attend to their storekeeping duties and this has become a recurring source of conflict among the members. It does not also help that there are bigger grocery stores nearby and in the town proper which negatively impacts the members' patronage of their community store. Moreover, most families have fairly stable sources of

income from employment and farming and that made them less dependent on the promise of profit that the store can bring to them. When asked about the prospect of improving the community store, an old resident came up and answered, "*Hindi uunlad yang tindahan kasi mayayaman ang mga taga-Dolores*" (Such store cannot be successful since Dolores' folks are financially well-off).

*Bahi* is a farming village in Gasan, Marinduque. Its main source of income is coconut and rice farming. Just seven kilometers off the town proper, the people of *Bahi* have access to larger grocery stores for their everyday use and consumption. Women mostly comprise its membership and men rarely participate in BEC-related activities. Its community store is usually empty and has rarely been replenished with products and goods due to lack of capital. The store has not gained adequate income due to members' bad credit standing and habit. Most of them would take the store items on sale but would delay their payments drying off the store's capacity to replenish its goods and resources. There seems to be a lack of concern from the people and the store has become more like a source of division and conflict among members. Attempts to remedy the situation have been initiated by the BPK leaders and foremost among them is to demand stricter regulations on the payment of credits but despite these attempts, a lack of commitment among the members affects the store's financial viability. Despite its generally conflicted management and supervision, there is a consensus among members to maintain the community store in their BPK.

### 3.2. Assessing the BPKs' Socio-Economic Integration

In this section, the paper attempts to evaluate the three (3) BPKs on how suitable each one of them is as a potential base for an economic dynamics that embodies the salient features of Jesus' compassion namely, the option for the poor (OP), the personalist/intersubjective/communion orientation (PICO), and the liberationist-enabling direction (LED). This evaluative attempt is largely based on the author's qualitative methodology that includes interview data, document review, and communal meetings and discussions. The basis of evaluation zeroes in on the extent of how a specific compassion attribute is manifested in each of the BPKs' integration of community stores. The evaluative rating renders the following numerical valuations: 3 (strongly demonstrated), 2 (moderately demonstrated), and 3 (limitedly demonstrated). Further description of the scale of demonstration is shown in Table 2 while the results of the evaluation are shown in Table 3.

**Table 2.** A demonstration scale.

| Demonstration Level | Score | Description |
|---|---|---|
| Strongly Demonstrated | 3 | The salient feature of Jesus' compassion narrative is fully exhibited by the BPK in its being and operations |
| Moderately Demonstrated | 2 | The salient feature of Jesus' compassion narrative is somewhat exhibited by the BPK in its being and operations |
| Limitedly Demonstrated | 1 | The salient feature of Jesus' compassion narrative is poorly exhibited by the BPK in its being and operations |

**Table 3.** Evaluative Scores of Three BECs relative to the salient features of Jesus' compassion narrative.

| BPK | Option for the Poor (OP) | Personalist/Intersubjecitve/ Communal Orientation (PICO) | Liberationist-Enabling Direction (LED) | Total |
|---|---|---|---|---|
| Panag Kalangkang | 3 | 3 | 3 | 9 |
| Dolores | 3 | 2 | 2 | 7 |
| Bahi | 2 | 1 | 1 | 4 |

*Panag Kalangkang* is a village community surrounded by the sea and when the sea is uncooperative, they resort to gathering together and finding a way to survive. Every member is limited in options due to geographical limitations. Their movement is restrained and the best course of action is for them to rely on each other as they could not afford to be self-interested, especially in the midst of restricted movement and limited resources. This contextual and situational reality explains why the community store is not just an alternative source of income and livelihood but also a community artifact of survival amid dearth and scarcity.

On the other hand, both *Dolores* and *Bahi*, owing to their geographical location and greater access to more resources, do not have the same urgency to rely on their community store even during weather disturbances since they live on the mainland. Aside from proximal access to larger stores for their basic goods, the members have jobs and employers that provide them regular income and salary. For them, the community store means nothing more than a place where goods are stored for purchase and consumption. It does not have that semiotic character that characterizes one's communal identity and belongingness. That lack or absence of self-identification with the store limits its potential as a space for increased interaction and interpersonal exchanges.

While the people of *Dolores* are determined to improve the current state of their community store following the success of *Panag Kalangkang*'s, they acknowledge the need to find a way to resolve those issues. The case of Bahi is entirely different. Most of them have shown indifference and to some extent, utter disregard, especially when some of them openly patronize other credit agencies or cooperatives instead of their own. The main complaint of most members is the sheer negligence of some of their members to pay the debts that they owe despite extended deadlines. According to them, what is rather more appalling is that these same people have been diligently paying their credit and dues in other credit agencies or cooperatives. In *Bahi*, more than blaming the geo-economic profile of the community as a reason for the store's failure, it is the lack or absence of a strong communal bond among its members resulting in weak foundational support for the community store to thrive and survive.

Despite the above-mentioned predicament, the paper assumes that BECs (or BPKs) can be a suitable base for compassion-based economics that is primarily person-oriented and relationship-based. Certain (pre)conditions are identified as strongly correlated with its success including communal cohesion and commitment, strong leadership and management, institutional support and guidance from the parish/diocese, and even its geo-economic profile/landscape.

## 4. Conclusions

Pope Francis asserts that amid the ill effects of the current economic system, there is a need for "new models of progress to arise . . . " (LS, 194). His "call for a new economic model is a call to empower local initiatives towards the formation and development of economic alternatives that... may at least offer a critical challenge to the hegemony of the dominant [economic system]" (Macaraan 2021, p. 77). The emphasis of Francis' call must not be situated between the binary matrix of the left-right ideology or the socialist-capitalist economic grand narratives but an appreciation of the value embodied in small, localized, grassroots-based communal economic alternatives. It is within this Francis' call where the paper hopes to specifically endorse the suitability of BECs to integrate a socio-economic component when implemented, exercised, and lived in real situations of small ecclesial communities. This paper argues that if only BECs would realize their vision of a Church of the Poor through the integral development of their members, the rationale for BEC's active socio-economic engagement would be therewith provided and would be sufficiently justified to challenge those with constant objections or apprehensions over an economically-inclined BEC dynamics.

It may even be logical (and even urgent) to push for a stronger and unyielding engagement of BEC into socio-economic agendum. Why not create and form (small)

neighborhood markets based on BEC structures in such a way that the substantivist values of kinship, religion, friendship, etc. are upheld more than the formalist values based on mere profit, utilities, techniques, etc.? With BEC's active socio-economic engagement (livelihood projects, cooperatives, small stores, etc.), as economic alternatives, what is created are not only projects but additional opportunities and scenarios for interpersonal and communal contact and network. In this way, a culture of fellowship and solidarity may even be more feasible due to these additional opportunities based on socio-economic interactions.

As found in field data, the success of socio-economic integration in BECs is founded on solid communal bonds as ecclesial communities. The communities need to first strengthen their communal dynamics before any attempt to incorporate any socio-economic initiative. These cases suggest that the success of a community store will rest on the strength of its communal makeup and its failure will generally be associated with weak communal ties as members of BEC.

**Funding:** This research was funded by the University Research Coordination Office (URCO) of the De La Salle University, grant number Project No. 18 N 1TAY14-3TAY14 and The APC was funded by the De La Salle University.

**Institutional Review Board Statement:** The study was conducted in accordance with the Declaration of Helsinki, and approved by the University Research Coordination Office (URCO) of the De La Salle University (Project No. 18 N 1TAY14-3TAY14, October 24, 2013).

**Informed Consent Statement:** Informed consent was obtained from all subjects involved in the study.

**Data Availability Statement:** The data that support the findings of this study are available from the corresponding author, [W.E.R. Macaraan], upon reasonable request.

**Conflicts of Interest:** The author declares no conflict of interest.

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
