# Peer review of "An Economic Alternative from the Base: The Basic Ecclesial Communities (BECs) in the Diocese of Boac, Philippines"

_religions, doi:10.3390/rel13070595_

Round 1

Reviewer 1 Report

1. Introduction
Para 3: "Utilizing Boff's... he suggests...". Did you mean to say "... it [i.e. the paper] suggests"? If so, then better to paraphrase the quoted part.
Para 4: "That market is a cultural..." seems incomplete
"It is argued that....that beyond or alongside.." Did you mean "... and that beyond..."?

The section mentions utilizing Boff, but moves to Nolan. Is mentioning Boff necessary at all?

2.Jesus' Narrative of Compassion

last para " section may be divided by subheadings." ?? error?

2.1 Nuancing...
para 3: "New Testament First was the word" Or "New Testament. First was the word"

2.2.1. Jesus identifies himself...
"A similar experience of relief and gratitude 
was felt by the sinners where after dining with Jesus in a table fellowship, it ended with the same call to live in peace" (please provide citation. Which event is being referred to here?)

2.2.3....
". Jesus’ non-violent means was however neither a matter of blind and passive submission to Roman oppression nor a violent uprising against it."
- If it is non-violent then "nor a violent" may be redundant

3.2. "lacks statistical treatment.." A statistical treatment could be done based on data from interviews and discussions.  

Descriptions in Table 2 are redundant (e.g. Strongly demonstrated = "strongly demonstrated...") and may be improved by making more specific and descriptive.

Author Response

First of all, I am grateful for the comments forwarded to me by your review and reading. I truly appreciate them and I agree that your comments have been of huge help in improving this paper. Below are my detailed responses. I have also improved the paper in terms of its grammar by using the grammarly app. Thank you.

  1. Response to First Reviewer's comment no.1a: paraphrased and clarified the use of Boff's dialectic hermeneutic. (p.1)
  2. Response to First Reviewer's comment no.1b: rewrote the sentence fragment as part of the previous sentence. (p.2)
  3. Response to First Reviewer's comment no.1c: The words "beyond and alongside" have been deleted and the sentence is improved. (p.2)
  4. Response to First Reviewer's comment no.2.1: added a period to read "New Testament. First" (p.3)
  5. Response to Reviewer 1 no.2.1..2: Added biblical reference for the statement. (p.4)
  6. Response to First Reviewer's comment no.2.2.3: Edited by removing the "non-violent" and replaced it with "proposed way was" (p.6)
  7. Response to First Reviewer's comment no.2.2.3a: Deleted the word "statistical treatment" to avoid confusion. (p.8)
  8. Response to First Reviewer's comment no.3.2b: Edited and improved the wording in the description part found in table 2 to avoid redundancy. (p.9)

In the end, I hope that my attempt to revise this paper is acceptable to you. Thank you again.

Reviewer 2 Report

This is an insightful piece of work. It combines valuable field research with an interesting conceptual framework. It has the potential to make a significant contribution to the debate on alternative economic structures.

At present the paper is marred by faults in presentation. Firstly it has grammatical faults. For example it is not always structured in sentences each of which has a verb. There are other errors, eg p3 para 2 'the dynamics.... was'. The paper needs to be read over and edited in detail by a reader known to the author with a good grasp of English grammar.

Secondly there are infelicities of style compared with the expectations in an academic journal. For example the sentence beginning 'Like many existing economic alternatives...' p2 para 2. Also 'right now' p2 para 2, 'rather bold' p6 para 3. p4 para 2 when Fox is introduced the reference should be immediate. Also the reference to a table far ahead p4 para 4 which will summarise a so far unstated structure.

If presentation is sharpened up (and with that perhaps a few details of the argument), I think this would make an excellent paper for the journal.

Theologically my main questions were

  • Is it valid to say that only this 'compassion based economics' sees a market as a social construct? Is that actually an insightful way of generalising across this and the others, solidarity, sharing, etc.?
  • The picture of Jesus on p6 sounds rather conveniently Liberal Kantian - universal solidarity, rights and duties - I wonder if those are rather unhistorical concepts to introduce esp. as you are priding yourself on a 'Before Christianity' reading

Author Response

First of all, I am grateful for the comments forwarded to me by your review and reading. I truly appreciate them and I agree that your comments have been of huge help in improving this paper. Below are my detailed responses. I have also improved the paper in terms of its grammar by using the grammarly app. Thank you. 

  1. Response to Second Reviewer's comment on the sentence "like many other economic alternatives...". The author decided to delete that sentence to address it since it may be misleading and confusing (see p. 2 of 11)
  2. Response to Second  Reviewer's comment on improving the grammar specifically on text "dynamics... was". The author has reread the paper and has revised some entries that contain grammatical errors and has also improved the sentences particularly those that contain fragments and basic subject-verb disagreements. Many of the grammatical errors that were corrected are also similarly raised by the other reviewer and I have addressed them too (see p.2 of 11)
  3. Response to Second Reviewer's comment on the use of "rather bold". The author rewrote the sentence (see p. 2 of 11)
  4. Response to Second Reviewer's comment on the immediate referencing to Fox. The author inserted the immediate referencing to Fox (see p. 3 of 11)
  5. Response to Second Reviewer's comment on the mention of Table 1 that is too far ahead of the table itself. The author decided to rewrite the sentence and add as well a sentence after the three features are detailed discussed to introduce the table 1 (see p. 4 of 11)

As for the theological questions that were raised, I consider the compassion-based economics as one of many economic alternatives that sees a social construct within the market. As stressed by Polanyi in his ideation of "Substantivist economics", he mentions how market in itself is not only rational but also social and is culturally-constructed or embedded. The reason why I use Nolan's "Before Christianity" is to provide a critical look into Jesus' praxis that has been popularly narrated over time not much in terms of "real" historical context but more on fiducial/faith context with strong theophilosophical embellishments. With Nolan's critical exposition of what really is meant by Jesus' words and deeds, there is a more nuanced approach to using the Gospel narrative of Jesus' praxis of compassion in relation with market dynamics and the values and ethos that must come with it.

In the end, I hope that my attempt to revise this paper is acceptable to you. Thank you again.

Round 2

Reviewer 1 Report

A number of grammatical improvements have been made. A little more improvement is necessary, which includes (though not limited to) the following:

Line 83. change "attempts" to "attempt"

Lines 106-108. "urged not....must be complemented" may be improved by using "not only... but also" e.g. "Authentic liberation involves not only a mere.... but also an audacious.."

Line 144. "The first was εςλεεω / εςλεαω 144 [eleeo / eleao], a verb..."

Line 149. "The second was the verb"

Line 152. "the verb"

[can omit the word "Greek"]

Line 176. "lists three salient" [omit (3)]

Line 410. "that they owe.."

Lines 410, 412. "debts to" instead of "credit and dues" may be more proper

Also,

Lines 272-273. "He believes..." [How do you know he believed that? Biblical citation, if any. Or is it a liberationist reading?] Is it Nolan's view? If so, please state it and critically evaluate it as well. 

Author Response

Dear Reviewer, 

Thank you for accepting my revisions based on your first comments. I truly appreciate it. 

As for the second set of revisions, the grammatical points for revisions have been duly noted and acted upon accordingly.

As for the last comment, I also have placed a direct quotation to reinforce the thought raised in the paper. I cited Nolan's work arguing that Jesus prophesies the fall of Israel if it enters into a war with the Romans.

Again my grateful appreciation for your time and effort in reading and commenting on this paper. I believe that your suggestions make the paper more scholarly readable and acceptable.

Reviewer 2 Report

I am grateful for the author's gracious response to my earlier comments.

The presentation of the paper has improved considerably since first submission.

There are now only a few minor points to be addressed. It is a great contribution to the debate on community-based economic models.

Author Response

Dear Reviewer, 

Thank you for accepting my revisions based on your first comments. I truly appreciate it. 

As for the second set of revisions, the grammatical points for revisions have been duly noted and acted upon accordingly.

As for comment no.1, the sentence is rewritten to address the ambiguity and possible confusion. It reads now as "In response to the call of Pope Francis (2019) "to set in place a new economic model", this paper hopes that compassion-based economics may be a feasible alternative to mainstream neoliberal economics."

As for comment no. 2, the sentence is also rewritten as "These cases suggest that the success of a community store will rest on the strength of its communal makeup and its failure will generally be associated with weak communal ties as members of BEC."

As for comment no. 3, about the inclusion of state, I completely agree that it will further nuance the argument but my concern is that since this is already located in the conclusion section, I had no solid or extended discussion of that aspect in the paper itself. But I will take note of that in subsequent papers since this paper can provide foundational points for what I consider the big project of compassion-based economics. 

Again my grateful appreciation for your time and effort in reading and commenting on this paper. I believe that your suggestions make the paper more scholarly readable and acceptable.
